# Prebiotic Peptide Bond Formation Through Amino Acid Phosphorylation. Insights from Quantum Chemical Simulations

**DOI:** 10.3390/life9030075

**Published:** 2019-09-16

**Authors:** Berta Martínez-Bachs, Albert Rimola

**Affiliations:** Department de Química, Universitat Autònoma de Barcelona, 08193 Bellaterra, Spain; berta.martinezi@e-campus.uab.cat

**Keywords:** prebiotic chemistry, biopolymers formation, condensation of amino acids, B3LYP-D3 calculations, clays, water solvent conditions

## Abstract

Condensation reactions between biomolecular building blocks are the main synthetic channels to build biopolymers. However, under highly diluted prebiotic conditions, condensations are thermodynamically hampered since they release water. Moreover, these reactions are also kinetically hindered as, in the absence of any catalyst, they present high activation energies. In living organisms, in the formation of peptides by condensation of amino acids, this issue is overcome by the participation of adenosine triphosphate (ATP), in which, previous to the condensation, phosphorylation of one of the reactants is carried out to convert it as an activated intermediate. In this work, we present for the first time results based on density functional theory (DFT) calculations on the peptide bond formation between two glycine (Gly) molecules adopting this phosphorylation-based mechanism considering a prebiotic context. Here, ATP has been modeled by a triphosphate (TP) component, and different scenarios have been considered: (i) gas-phase conditions, (ii) in the presence of a Mg^2+^ ion available within the layer of clays, and (iii) in the presence of a Mg^2+^ ion in watery environments. For all of them, the free energy profiles have been fully characterized. Energetics derived from the quantum chemical calculations indicate that none of the processes seem to be feasible in the prebiotic context. In scenarios (i) and (ii), the reactions are inhibited due to unfavorable thermodynamics associated with the formation of high energy intermediates, while in scenario (iii), the reaction is inhibited due to the high free energy barrier associated with the condensation reactions. As a final consideration, the role of clays in this TP-mediated peptide bond formation route is advocated, since the interaction of the phosphorylated intermediate with the internal clay surfaces could well favor the reaction free energies.

## 1. Introduction

Formation of biopolymers was one of the crucial steps in the sequence of organizational events that led to the origin of life on earth [1,2,3,4,5,6,7,8,9]. Peptides, polynucleotides, and polymeric carbohydrates are considered essential biopolymers for life. Interestingly, in most cases, their formation takes place through condensation reactions, which join the corresponding biomolecular building block precursors, namely monomers. For instance, in the case of peptides, amino acids condense through a peptide bond that links them (Figure 1A). In the case of a dinucleotide, three condensation reactions are required. First with the formation of an N-glycosidic bond between a nitrogenous base (NB) and a ribose sugar (R) to generate a nucleoside (NS) as follows NB + R → NS + H_2_O (Figure 1B). Then a reaction between NS and phosphate (Pi) via a phosphoester bond with the formation of a nucleotide (NT): NS + Pi → NT + H_2_O (Figure 1C). Finally, two nucleotides react to form a dinucleotide (dNT), which is being held together with a phosphodiester bond: NT1 + NT2 → dNT + H_2_O (Figure 1D). It should be noted that during the formation of such a dinucleotide, three molecules of water are generated.

Therefore, condensation reactions form chemical bonds that join the molecular building blocks. However, this is achieved at an expenditure of energy and the generation of one water molecule per condensation reaction (Figure 1). In the prebiotic context, such condensation reactions that lead to the formation of biopolymers present major challenges. That is, these reactions are thermodynamically unfavorable in water, which is a plausible situation since reactants might well have been diluted in primordial oceans. Experimental thermodynamic measurements do indeed indicate that these reactions are disfavored in watery environments. If the condensation reaction is written as in Equation (1)
X (solution) + Y (solution) ⇄ X-Y (solution) + H_2_O,(1)
where X and Y are generic monomers, the free energy (Δ*G*) of dimerization is positive: + 14.25 kJ moL^−1^ at normal conditions for X = Y = glycine [10], and + 17.3 kJ moL^−1^ at 37 °C for X = glycine and Y = alanine [5]. According to that, we are faced with the water paradox [11]: Water is essential for life’s emergence [12], but in this respect water is a hindrance to the emerging chemical evolution of life.

A set of well-established and extended theories devoted to overcoming the thermodynamic problem are those that advocate minerals as cradles of the first biopolymers [13,14,15,16,17]. The underlying idea is that mineral surfaces present specific sites capable of concentrating and activating the prebiotic monomers, and even capture the released water, thereby favoring the condensation reactions overall [18,19]. A large number of experiments have been carried out supporting the validity of the “on the rocks” approach. They have been particularly devoted to the condensation of amino acids to form peptides in the presence of different minerals (e.g., silica [20,21,22], TiO_2_ [20,21,23,24], alumina [25,26,27,28], clays [16,29], among others [30,31,32,33]) and to the elongation of nucleotides in clays [34]. Additionally, several theoretical works also simulated these processes, in most of the cases indicating that the presence of minerals not only favor the reactions thermodynamically but also catalyze them by lowering the free energy barriers. Reviews on the theoretical works devoted to these kinds of processes have recently appeared in the literature [35,36,37,38].

Despite these positive assessments, the present work aims to tackle the biopolymer formation problem from a different perspective, which is inspired by the mechanisms carried out by the biological machinery of living organisms. A living organism is capable of synthesizing their own biopolymers, in which in some cases adenosine triphosphate (ATP) is an essential component as it is used as an energy source to make the overall reaction thermodynamically favorable. The general process involves, first, the reaction of the terminal phosphate group of ATP with a carboxylic acid group (e.g., the C-terminus of a peptide) to form an activated phosphoric carboxylate anhydride, and then, the reaction of this activated species with an alcohol, thiol or amino groups to condensate the carboxylate groups with them (i.e., forming the corresponding ester, tioester, and amide bonds, respectively). In the present work, this ATP-mediated mechanism has been investigated for the peptide bond formation between two glycine molecules as a reaction test case study using quantum chemical calculations. Figure 2 shows the steps for this reaction.

The first step involves the activation of the carboxyl group of one of the reactants by ATP (i.e., phosphorylation through its polyphosphate chain), in which a phosphoester bond is formed, and an adenosine diphosphate (ADP) molecule is released. In the second step, this activated intermediate reacts with the other reactant, in which the actual peptide bond is formed, followed by the elimination of a phosphate (Pi).

Since the chemically active part of ATP is the triphosphate chain, we considered here the reaction to be mediated by the terminal high energy phosphate of the triphosphate group. This scenario, moreover, is plausible considering prebiotic conditions, since the triphosphate might have well been present dangling from the clay surface and the fact that chemical nucleoside is not necessarily a requirement during the primordial conditions on the early earth. Furthermore, the ATP molecule may not have been available because the molecule is highly complex, as well as being highly unstable and readily deteriorates. The most important point is that a triphosphate molecule alone will suffice as a choice of energy bearing molecule. Moreover, there is clear evidence that the meteoritic mineral schreibersite, (Fe, Ni)_3_P, with water, produces HPO_3_^2−^ and H_2_PO_2_^−^ as the major phosphorus-based anions [39], whose oxidation, in turn, leads to the formation of condensed phosphates (such as triphosphates) [40]. Remarkably, these oxidation reactions could have indeed been plausible under early Precambrian conditions (i.e., the conditions on earth at 3.0 billion years ago). It is worth mentioning that different experimental works have successfully demonstrated the role of phosphorylation reactions in the synthesis of relevant molecular precursors of prebiotic interest [41,42,43,44].

With the aim to investigate the plausibility of the proposed phosphorylation-based mechanism in the prebiotic context in depth, three different scenarios have been considered: (i) under gas-phase conditions, (ii) in the presence of Mg^2+^ ions within the layers of clays, and (iii) in the presence of Mg^2+^ ions in water. The work aims to assess whether the reaction energetics provided by the theoretical simulations are actually favorable, hence opening up new routes to studying the chemical evolution of essential biopolymers on the early earth.

## 2. Methods

All calculations were carried out with the Gaussian09 package programs [45]. Geometry optimizations were performed using the density functional theory (DFT)-based hybrid three-parameter B3LYP method [46,47], which was combined with the posterior Grimme’s correction D3 term [48] to account for dispersion forces missing in the pure B3LYP method. As a basis set, the Pople’s 6-311++G(d,p) one was used. The nature of all the stationary points has been characterized by the analytical calculation of respective harmonic frequencies as minima (reactants, products, and intermediates, with no imaginary frequencies) and saddle points (transition states, with one imaginary frequency, whose eigenvector is associated with the reaction path). In some intriguing cases, intrinsic reaction coordinate (IRC) calculations were performed to ensure that a given transition state connects with the corresponding minima of the potential energy surfaces. To obtain free energy values, thermochemical corrections to the potential energy values were computed using the standard statistical thermodynamics formulae based on the partition functions derived from the rigid/rotor harmonic oscillator approximations [49]. 

## 3. Results and Discussion

As mentioned in the Introduction, the peptide bond formation is a condensation reaction between two amino acids. Under strict gas-phase conditions, in which both amino acids are in their canonical form (i.e., NH_2_CHRCOOH, R being the side chain), the reaction mechanism is concerted and involves two simultaneous chemical processes: (i) a nucleophilic reaction of the N atom of one amino acid to the carboxyl C atom of the other amino acid, hence forming the N-C bond, and (ii) a proton transfer from the amino (NH_2_) group of the first amino acid to the hydroxyl (OH) group of the second amino acid. In the process, a water molecule is generated. In the case of the condensation of two glycine (Gly) molecules, the reaction has a free energy barrier at T = 298 K (Δ*G*_298_^≠^) of 41.8 kcal/moL and a reaction free energy (Δ_r_*G*_298_^0^) of −3.5 kcal/moL (see Figure 3). It is worth mentioning that this gas-phase process is physically unsound in a prebiotic context, but the calculated energetics will be useful here to assess the role of triphosphate in the intrinsic energetics. In this work, the peptide bond formation between two Gly in the presence of triphosphate (TP) has been simulated adopting different conditions. In Section 3.1, the TP-mediated reaction in the gas phase is presented to check the intrinsic effect of TP in the reaction energetics. In Section 3.2, the focus is on the reaction in the presence of a Mg^2+^ ion in dry conditions, this way resembling plausible conditions present within the internal surfaces of clays. Finally, in Section 3.3, results on the reaction in the presence of a Mg^2+^ ion in watery environments are shown.

### 3.1. TP-Mediated Peptide Bond Formation in the Gas Phase

Figure 4 shows the free energy profiles of the peptide bond formation reaction due to the condensation of two Gly molecules in the presence of TP in the gas phase. Due to the strict dry conditions (i.e., absence of water), both Gly molecules and TP are in their canonical forms, namely, Gly as NH_2_CH_2_COOH and TP as a triphosphoric acid (H_5_O_10_P_3_).

The reaction has been computed to occur in two steps (shown in Figure 4A,B). The first step is the formation of the glycylmonophosphate (GMP) as intermediate. The reactants for this step are one Gly molecule and TP (see Figure 4A). Formation of GMP occurs in a concerted way, in which Gly performs an O-nucleophilic reaction to the P atom of one of the TP ends (hence forming a phosphoester bond), followed by a proton transfer from the Gly OH group to the O atom of the phosphate bond. The O-nucleophilic reaction can be carried out by either the OH or the CO Gly groups. In the former case, the reaction takes place through a highly-strained 4th-membered ring transition state structure (see TS1-gp of Figure 4A), while in the later through a less strained 6th-membered ring one (see TS2-gp of Figure 4A). Accordingly, calculated Δ*G*_298_^≠^ values are 56.0 and 47.8 kcal moL^−1^, respectively. As a result of the process, a phosphate bond is broken, thus forming GMP and releasing diphosphoric acid. The calculated Δ_r_*G*_298_^0^ is + 11.7 kcal moL^−1^, namely, endergonic, due to the rupture of the highly energetic phosphate bond. It is worth mentioning that here we did not consider the formation of a pre-reactant complex (namely, the reactants interacting between them, forming a super-structure), since we assume that in the gas-phase conditions, Gly approaches to TP to directly react with it. The same assumption has been considered for the subsequent step described below.

The second step is the actual peptide bond formation, in this case, the condensation between GMP with a second Gly molecule (see Figure 4B). The adopted mechanism is very similar to that between two pure Gly molecules, i.e., a concerted process in which the N atom of the Gly molecule nucleophillically reacts with the C atom of GMP and a proton transfer from the NH_2_ group of Gly to the phosphate moiety. However, the proton receiver can be the O atom belonging to the phosphoester bond of GMP or the terminal O atom, forming 4th- and 6th-membered ring transition states (see TS3-gp and TS4-gp, respectively). Because of the lower geometrical constraint of the 6th-membered ring structure than the 4th-membered ring one, the former has a lower Δ*G*_298_^≠^ value than the later (22.1 and 24.2 kcal moL^−1^, respectively). In both cases, the product is the same, the glycylglycine (Gly-Gly) molecule and the release of phosphoric acid, with Δ_r_*G*_298_^0^ = −7.5 kcal moL^−1^ (both values with respect to the Gly + GMP asymptote, AS3-gp).

According to these results, it seems clear that the reaction presents favorable energetics when one of the two Gly molecules is activated by GMP, which has a moderate free energy barrier and a negative reaction free energy. However, the critical point is the formation of this activated GMP form, as it has largely disfavored energetics, both kinetically and thermodynamically. Due to this disfavored first step, the energetics of the second step relative to the “2Gly + TP” general asymptote become Δ*G*_298_^≠^ = 33.8 kcal moL^−1^ (too high to proceed at normal conditions) and Δ_r_*G*_298_^0^ = + 4.2 kcal moL^−1^ (an endergonic reaction).

### 3.2. TP-Mediated Peptide Bond Formation in the Presence of A Mg^2+^-Clay Cation

As mentioned in the introduction, condensation reactions in the presence of minerals have been demonstrated experimentally and theoretically to be actually effective. Among these minerals, clays play a central role, as they are very active in different condensation reactions of prebiotic interest. Clays are Al-containing phyllosilicates presenting a multi-layered structure, in which one layer consists of [SiO_4_] tetrahedral silicate sheets and the other octahedral sheets of [AlO_6_]. Isomorphic substitutions of Si and Al by di/trivalent metal cations are recurring so that the negative charge generated is compensated by the presence of other metal cations (e.g., Li^+^, Mg^2+^) in the interlayer regions, which interact with the sheets. In this section, we investigate the TP-mediated peptide bond formation in the presence of a Mg^2+^ ion, which interacts with TP and is surrounded by four H_2_O molecules as a way to resemble a Mg^2+^ single site present within the interlayer regions of clays (hereafter referred to as Mg^2+^-clay). Thus, the four H_2_O molecules do not represent hydration conditions but the first sphere around Mg^2+^ due to its interaction with the internal clay sheets.

As for the gas-phase case, the overall TP-mediated reaction involves two steps: (i) formation of GMP (in this case attached to Mg^2+^-clay, hereafter referred to as Mg-GMP), and (ii) formation of the peptide bond. The calculated free energy profiles are shown in Figure 5A,B, respectively.

For the formation of the Mg-GMP intermediate (see Figure 5A), the initial reactants are one Gly molecule and triphosphoric acid. Here, we consider that the interlayer regions of the clays present moderately dry conditions, and accordingly, Gly is in its canonical form. In relation to triphosphoric acid, one of its ends attaches to Mg^2+^-clay through two O atoms, which, to keep the system electroneutral, are deprotonated (see AS1-clay of Figure 5A).

The actual formation of Mg-GMP takes place by an O-nucleophilic reaction of Gly to the P atom followed by the proton transfer. Here, for the sake of clarity, we only focused on the nucleophilic reaction carried out by the carbonyl (C = O) functional group of Gly, as it forms a 6th-membered ring transition state structure (TS1-clay) with Δ*G*_298_^≠^ = 42.1 kcal moL^−1^. Interestingly, the presence of Mg^2+^-clay induces an energy barrier decrease of *ca.* 6 kcal moL^−1^ in comparison to the same process in the absence of Mg^2+^-clay (for TS2-gp, Δ*G*_298_^≠^ = 47.8 kcal moL^−1^, see Figure 4A). Such a decrease is because the interaction of TP with Mg^2+^-clay exerts an increase of the P−O bond polarization, making the P atom more electrophilic, thus favoring a phosphoester bond formation. However, the calculated Δ_r_*G*_298_^0^ value to form Mg-GMP + diphosphoric acid (see AS2-clay) is 21.8 kcal moL^−1^, indicating that this step is a thermodynamically unfavorable process.

The second part of the reaction involves the condensation of a second (canonical) Gly molecule with Mg-GMP (see Figure 5B). This is carried out in two steps. The first step is the formation of the N−C bond through the nucleophilic reaction of the second Gly to Mg-GMP (see TS2-clay), presenting a calculated Δ*G*_298_^≠^ of 18.6 kcal moL^−1^. The resulting structure is an intermediate species (I-Mg) containing a zwitterionic moiety, i.e., NH_2_CH_2_CO (Pi)O(^−^)NH_2_(^+^)CH_2_COOH, which is stable (13.5 kcal moL^−1^ above the AS3-clay asymptote) due to the charge stabilizing effects conferred by its interaction with Mg^2+^-clay and the phosphate. That is, the positively charged NH_2_(^+^) group H-bonds to the negatively charged [O] atom of the phosphate group. The second step consists of a proton transfer from the NH_2_(^+^) group to the closest P-O phosphate group (see TS3-clay) to form finally Gly-Gly with a calculated Δ*G*_298_^≠^ = 18.4 kcal moL^−1^. The calculated Δ_r_*G*_298_^0^ of this second step, −6.4 kcal moL^−1^ from the AS3-clay asymptote, indicates that it is exergonic. Interestingly, both energy barriers to form Gly-Gly from Mg-GMP are similar and lower (about 18.5 kcal moL^−1^) than that in the absence of Mg^2+^-clay (22.1 kcal moL^−1^), indicating that the presence of the metal cation exerts catalytic effects, probably due to C = O bond polarizing effects.

According to this energetic data, and similarly to what was observed in the gas phase, formation of Mg-GMP as an activated complex is crucial for the subsequent peptide bond formation, since from Mg-GMP the condensation presents favorable energetics, i.e., moderate free energy barriers (Δ*G*_298_^≠^ ≈ 18.5 kcal moL^−1^) and exergonic characteristics. Nevertheless, the most energetically demanding step is the Mg-GMP formation, as it has a high energy barrier (Δ*G*_298_^≠^ ≈ 42.1 kcal moL^−1^) and is endergonic (Δ_r_*G*_298_^0^ = 21.8 kcal moL^−1^). This endergonicity, however, exists by considering the energy differences between the two asymptotes, namely, AS1-clay and AS2-clay, but one has to keep in mind that the reactions are occurring within layers of the clay itself. In this sense, it is important to mention that we omitted typical but crucial steps in surface reactions, that is: (i) the diffusion of the reacting components in the interlayer regions to approach and be in close proximity between them, and (ii) the interaction of the products with the internal clay sheets. Both steps were omitted due to the limitation of our model that represents the clay. Since our focus is on elucidating the effects of a Mg^2+^ ion as a single Lewis site present in the clay sheets, we only consider the Mg^2+^ ion and its surroundings, this later aspect by coordinating the Mg^2+^ ion with four water molecules.

Thus, neither the actual clay sheets nor the interlayer regions have been considered in our models and accordingly neither the diffusivity of the reactants nor the interaction of the products with the clay sheets can be simulated. Thus, it is worth mentioning that consideration of these two steps can significantly affect the energetics of the global process. That is, the interaction of the reactants with the clay sheets and their rate of diffusion introduces additional intermolecular forces to overcome, probably increasing the free energy barriers. On the other hand, the interaction of the products with the clay sheets introduces stabilizing effects that could favor the reaction, probably decreasing its free energy, even to a negative value. Thus, more investigations are required, using more realistic clay structural models (i.e., periodic systems), to have conclusive energetics and to know how favorable the processes are.

### 3.3. TP-Mediated Peptide Bond Formation in the Presence of a Mg^2+^ ion and in Watery Environments

In this final section, the condensation between two Gly molecules has been investigated in the presence of a Mg^2+^ ion in water. This implies that Mg^2+^ is actually solvated, that is, at variance with the previous section, water molecules coordinating the Mg^2+^ ion are not mimicking the clay sheets but do indeed constitute the first solvation coordination sphere. In addition to that, a direct consequence of the water solvation environment is that Gly molecules have to be in their zwitterionic states (i.e., ^+^H_3_NCH_2_COO−). It was long reported that five water molecules are enough to largely stabilize zwitterions compared to the canonical systems [50,51]. Therefore, we simulate the water environment with five water molecules per Gly component (in addition to those present in the Mg^2+^ first coordination sphere).

Figure 6A shows the free energy profile for the formation of the GMP intermediate (in this section, hereafter referred to as Mg-GMP-hydr). The pre-reactant complex, R1-hydr, has a Gly molecule, which is in close proximity to the TP interacting via H-bonds. In contrast to the previous sections, here we consider the R1-hydr complex as a relevant structure in our calculated free energy profile because, previous to the reaction, the reactants have to be in close proximity in a fully solvated way forming a stable super-structure. Since R1-hydr is more stable than the AS1-hydr asymptote, R1-hydr is taken to be as the energy reference point for this first step. Since Gly is in its zwitterionic form, Mg-GMP-hydr formation occurs through the O reaction of the Gly COO^−^ group towards a P atom of TP, followed by the breaking of a P-O bond of TP to release diphosphoric acid (see TS1-hydr), with Δ*G*_298_^≠^ = 30.4 kcal moL^−1^. It is worth mentioning that, despite the stabilizing effects introduced by the solvation, in which the pre-reactant R1-hydr is a stable structure, this calculated energy barrier is the lowest one leading to the formation of GMP compared to the other two analog-processes. Such a decrease is probably due to the zwitterionic state of Gly, in which the negatively charged COO^−^ group makes the nucleophilic reaction to the P atom more effective.

The formed I1-hydr structure (Figure 6A) contains the activated Mg-GMP-hydr intermediate, in which the Gly component presents the amino group as NH_3_^+^, while the released diphosphoric acid has one deprotonated P-O group, hence balancing the overall neutral charge (see I1-hydr). The I1-hydr structure is likely a species ready to react with the second Gly to form the Gly-Gly. However, to save computational time for the subsequent peptide bond formation, we removed the diphosphoric acid from the medium, and this implies to convert the positively charged Gly moiety into neutral, i.e., NH_3_^+^ into NH_2_. To this aim, we carried out a proton transfer from NH_3_^+^ to the negatively PO^−^ group, which is assisted by one water molecule (see TS2-hydr). The Mg-GMP-hydr product becomes fully neutral as well as the diphosphoric acid (see P1-hydr). Interestingly, I1-hydr and P1-hydr are close in energy, the former being more stable than the later by 1.6 kcal moL^−1^, whose conversion has Δ*G*_298_^≠^ = 16.6 kcal moL^−1^. For both structures, the reaction free energies for their formation are largely favorable (*cf*. Δ_r_*G*_298_^0^ = −22.5 and −20.9 kcal moL^−1^, respectively).

The peptide bond formation by reaction of Mg-GMP-hydr with the second Gly molecule (surrounded by five water molecules and being in its zwitterionic form) is shown in Figure 6B. In this step, we also accounted for a pre-reactant complex (R2-hydr) since it is at −13.9 kcal moL^−1^ more stable than the AS2-hydr asymptote. In R2-hydr, the incoming second Gly molecule is in close proximity to Mg-GMP-hydr, both structures being engaged by H-bond interactions mediated by the water molecules. A critical point for this condensation reaction is that the second incoming Gly molecule is a zwitterion ion and, accordingly, the N atom is not able to carry out the nucleophilic reaction because no lone pairs are available. Thus, the first step to take into account is the conversion of the inactive NH_3_^+^ group into an active NH_2_ one. This is achieved by transferring one of the protons of NH_3_^+^ to one of the available PO groups. In our case, such a transfer is assisted by two water molecules (see TS3-hydr), with Δ*G*_298_^≠^ of 9.2 kcal moL^−1^. The resulting intermediate species (I2-hydr) is at 6.8 kcal moL^−1^ more unstable compared to the R2-hydr complex. I2-hydr is now an active species to proceed with the peptide bond formation. This takes place by the nucleophilic reaction to form an N-C bond by transferring a proton from the NH_2_ to an available PO group, followed by the breaking of the C-O(P) bond and then releasing of the Gly-Gly (see TS4-hydr). This transition state has Δ*G*_298_^≠^ = 40.5 kcal moL^−1^, very high to proceed at normal conditions. It is worth mentioning that TS4-hydr presents a 4th-membered ring, which differs from the 6th-membered rings present in TS3-clay and TS3-gp. This is because the TS4-hydr complex has a uniquely placed P-O bond to receive a proton from a phosphoester bond; the other PO groups are either interacting with Mg^2+^-hydr or are already protonated (because of the previous step, TS3-hydr). Despite the high energy barrier shown by TS4-hydr, the intrinsic Δ_r_*G*_298_^0^ value was calculated to be at −19.4 kcal moL^−1^, indicating that the process is, at least, thermodynamically favorable.

As a general consideration of this process, it is worth mentioning that both steps present highly favorable reaction free energies due to the stability of the products in water. That is, both processes have been identified as exergonic, with noticeable negative reaction free energies. The main energetic limitation of this process is the actual peptide bond formation between Mg-GMP-hydr and Gly, as it has Δ*G*_298_^≠^ = 40.5 kcal moL^−1^. This high energy barrier is due to the large stability of the pre-reactant R2-hydr, which is a consequence of the solvation effects, i.e., the H-bonds between the two Gly/Gly reactants and the water molecules.

## 4. Conclusions

The peptide bond formation reaction through condensation of two glycine (Gly) molecules was investigated using quantum chemical calculations at the B3LYP-D3/6-311++G(d,p) theory level. The adopted mechanism consisted of two steps, namely phosphorylation of one Gly via reaction with the energetic triphosphate (TP) to form an activated intermediate; and the condensation of the formed intermediate with the second Gly molecule to arrive at the glycylglycine (Gly-Gly) dipeptide. The inspiration for this mechanism was drawn from biology since it adopts a similar mechanism for condensation reactions carried out in living organisms, in which the phosphorylation source is ATP. With the aim of confirming for the first time the plausibility of the proposed mechanisms in a prebiotic context, three different scenarios were considered: (i) reactions in gas phase, (ii) during the presence of Mg^2+^ ions within the interlayer regions of clays (Mg^2+^-clay), and (iii) in the presence of Mg^2+^ ions in water (Mg^2+^-hydr). The main results are summarized as follows:

Condensation of the phosphorylated intermediate with Gly molecules was found to be thermodynamically favorable during all three scenarios. However, under gas-phase conditions and in the presence of Mg^2+^-clay, the formation of the phosphorylated intermediates showed large exergonicity, disfavoring the overall process from a thermodynamic standpoint. In contrast, in the presence of Mg^2+^-hydr complexes, the phosphorylation reactions were also thermodynamically favorable due to the stabilizing effects of the water.

The free energy barriers for the formation of the activated intermediate showed a trend, from high to low: gas-phase > Mg^2+^-clay > Mg^2+^-hydr. The interaction between the Mg^2+^ ions and the triphosphates polarized the P-O bonds and this, in turn, made phosphorous atoms more prone to react with the incoming second Gly. In water, since Gly is in its zwitterionic state, the O_(Gly)_−P_(TP)_ bond formation is more effective because of the charge of the COO−_(Gly)_ group.

In the gas phase and in the presence of Mg^2+^-clay free energy barriers associated with the condensation reactions are moderately high, between 18 and 22 kcal moL^−1^. This is, however, not the case with Mg^2+^-hydr complex, as the free energy barrier corresponding to the actual peptide bond formation is about 40 kcal moL^−1^, thus the reaction is kinetically hampered. The reason for this high value arises from the enhanced stability of the pre-reactant structure because of solvation effects.

In summary, none of the processes studied under different scenarios seem to be feasible in the prebiotic context. Both in the gas-phase and with Mg^2+^-clay complexes, the reactions are inhibited due to unfavorable thermodynamics as a result of the formation of stubborn associated intermediates; with Mg^2+^-hydr the reaction is inhibited due to the high free energy barrier associated with the condensation reactions. Despite some negative results, it is worth making a final note relevant to the reactions occurring within the clay interlayers as follows: We only considered single Lewis structure sites within the layers, but it should be obvious that full merits of the clay sheets should also be taken into account. As postulated by the “on the rocks” approach, the interaction of the intermediates and/or of products with the internal clay surfaces can induce a non-depreciable stabilization of them, hence favoring the reactions. This stabilizing attribute, due to the limitations of our structural modeling (i.e., only a Mg^2+^ ion surrounded by four H_2_O molecules), was not fully explored in this work. It is possible that the catalytic effects exerted by a single Mg^2+^ ion site, alongside the stabilizing effects of the intermediates, resultant products, together with the clay-surface interactions could highly favor energetical reactions during the overall processes. This is a promising hypothesis that deserves to be investigated in forthcoming works.

## Figures and Tables

**Figure 1 life-09-00075-f001:**
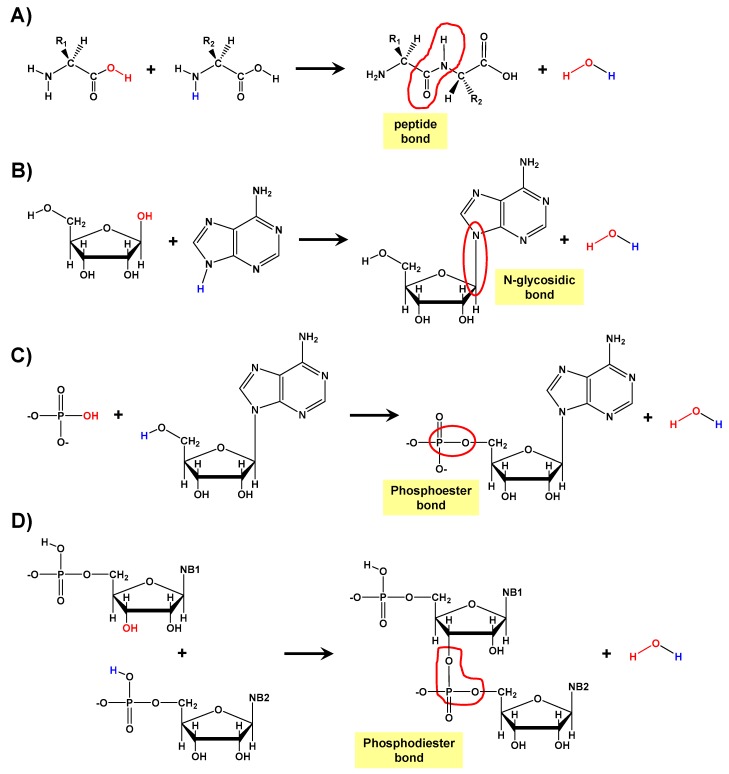
Different condensation reactions relevant for the formation of biopolymers: (**A**) between two amino acids forming a dipeptide, (**B**) between ribose and a nitrogenous base forming a nucleoside, (**C**) between the nucleoside and a phosphate forming a nucleotide, and (**D**) between two nucleotides forming a dinucleotide. In all these condensation reactions, water molecules are released, whose colors indicate the origin of each atom.

**Figure 2 life-09-00075-f002:**
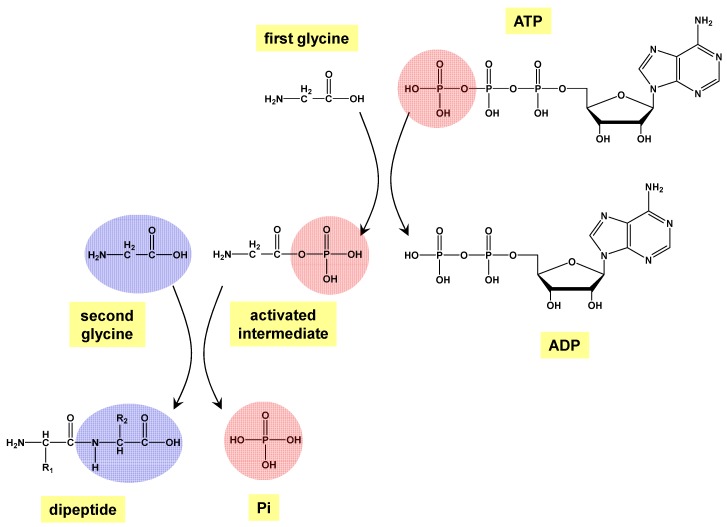
Processes involved in the peptide bond formation reaction between two glycine molecules mediated by adenosine triphosphate (ATP) as studied in this work. The first step is the formation of an activated intermediate (phosphoglycine) during the reaction between the carboxylic (-COOH) group of glycine and the terminal high energy phosphate of the ATP molecule; in the process, the latter is converted to adenosine diphosphate (ADP). The second step is the condensation between the activated intermediate and a second glycine molecule forming the peptide bond and releasing a phosphate. Note that for the sake of clarity, the process is shown in its neutral/canonical form, i.e., without formal charges and without protonated/deprotonated states.

**Figure 3 life-09-00075-f003:**
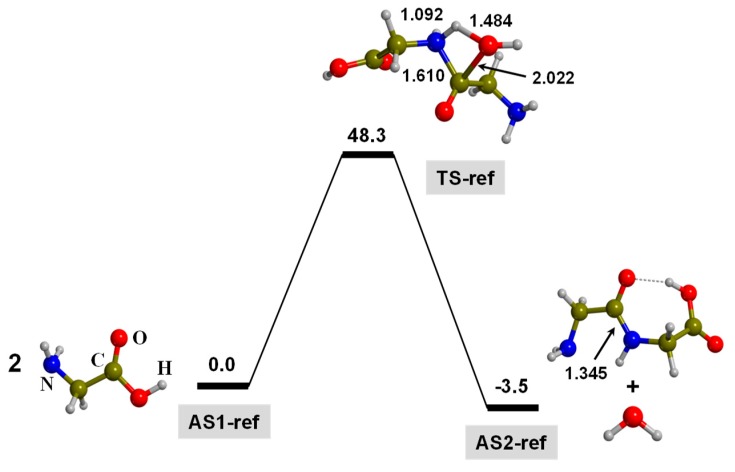
B3LYP-D3/6-311++G(d,p) free energy profile (in kcal moL^−1^) at T = 298 K for the condensation of two glycine molecules in the gas phase. Relative energies are with respect to two glycine (Gly) molecules infinitely separated (AS1-ref reference state). Bond distances are in Å. Pure potential energy values and including zero-point energy corrections are provided as Appendix A.

**Figure 4 life-09-00075-f004:**
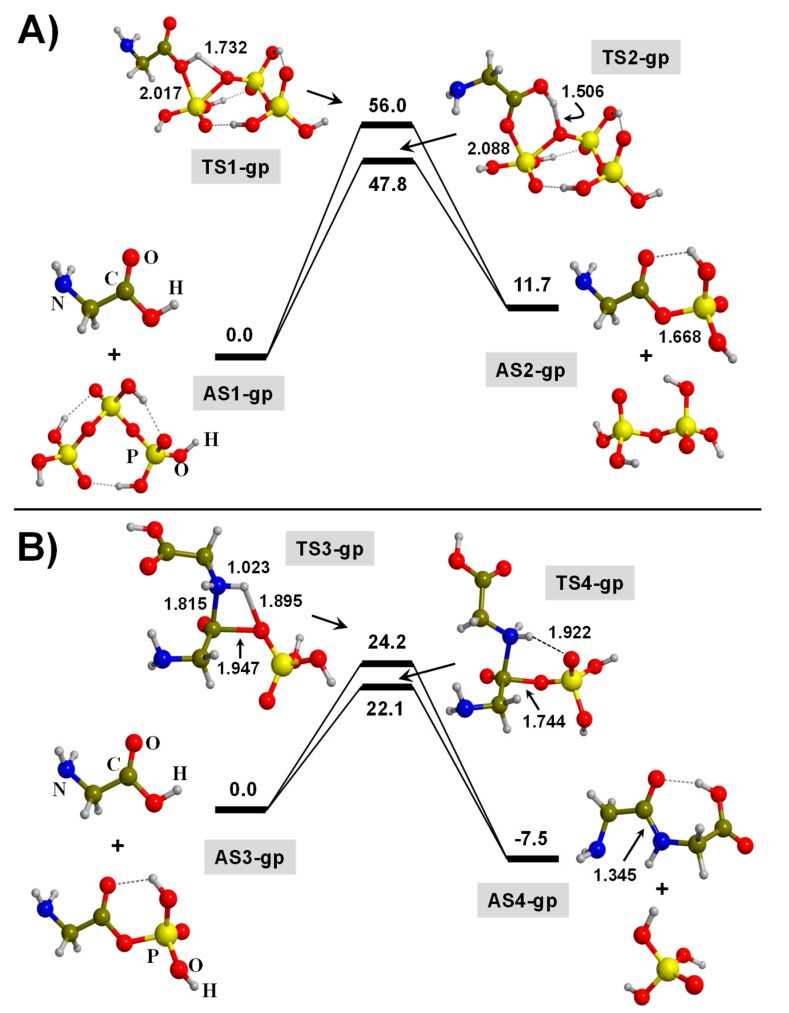
B3LYP-D3/6-311++G(d,p) free energy profiles (in kcal/moL) at T = 298 K for the triphosphate (TP)-mediated condensation of two glycine molecules in the gas phase: (**A**) formation of the glycylmonophosphate (GMP) activated intermediate; (**B**) formation of the peptide bond between GMP and a second glycine molecule. Relative energies are with respect to the corresponding AS1-gp and AS3-gp reference states. Bond distances are in Å. Pure potential energy values and including zero-point energy corrections are provided as Appendix A.

**Figure 5 life-09-00075-f005:**
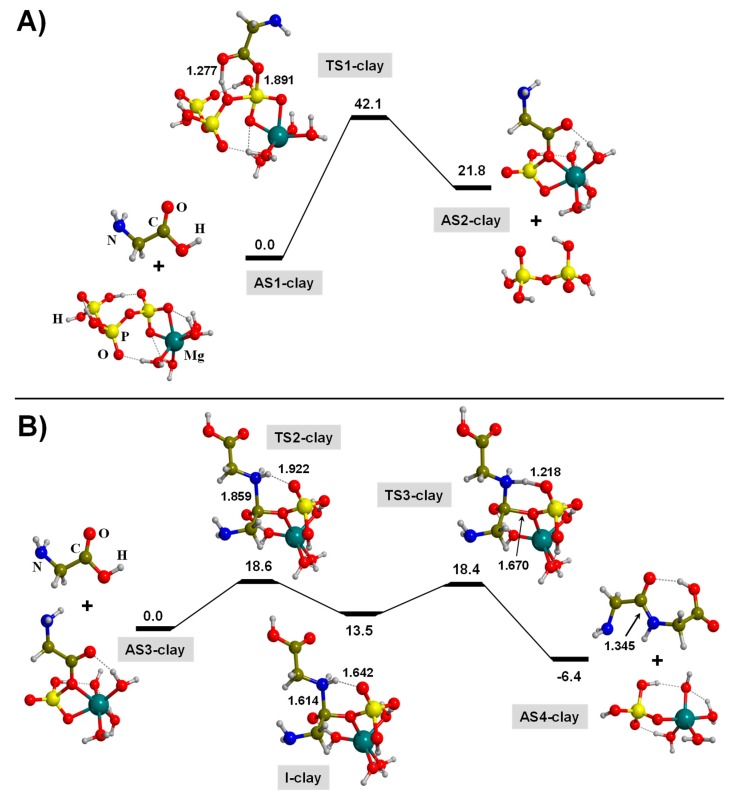
B3LYP-D3/6-311++G(d,p) free energy profiles (in kcal moL^−1^) at T = 298 K for the TP-mediated condensation of two glycine molecules in the presence of Mg^2+^-clay: (**A**) formation of the Mg-GMP activated intermediate; (**B**) formation of the peptide bond between Mg-GMP and a second glycine molecule. Relative energies are with respect to the corresponding AS1-clay and AS3-clay reference states. Bond distances are in Å. Pure potential energy values and including zero-point energy corrections are provided as Appendix A.

**Figure 6 life-09-00075-f006:**
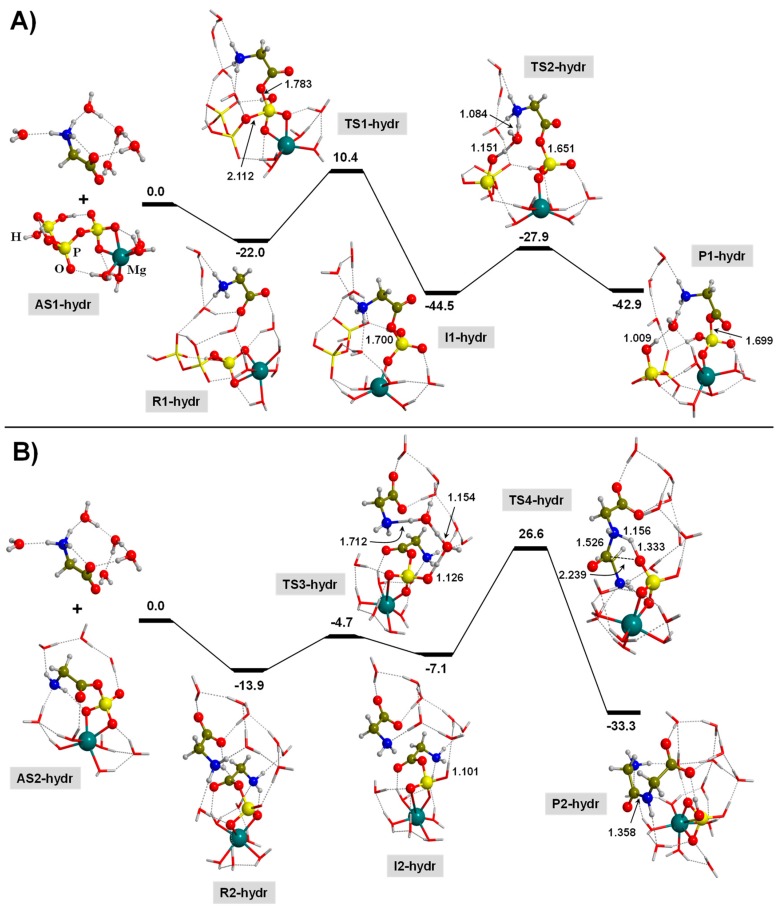
B3LYP-D3/6-311++G(d,p) free energy profiles (in kcal moL^−1^) at T = 298 K for the TP-mediated condensation of two glycine molecules in the presence of a Mg^2+^ ion and in the presence of water solvent molecules: (**A**) formation of the Mg-GMP-hydr activated intermediate; (**B**) formation of the peptide bond between Mg-GMP-hydr and a second glycine molecule. Relative energies are with respect to the corresponding AS1-hydr and AS2-hydr reference states. Bond distances are in Å. Pure potential energy values and including zero-point energy corrections are provided as Appendix A.

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
