# Peer review of "Prebiotic Peptide Bond Formation Through Amino Acid Phosphorylation. Insights from Quantum Chemical Simulations"

_life, 2019, doi:10.3390/life9030075_

Round 1
Reviewer 1 Report
Bioinspired synthesis of prebiotic molecules is a new trend in prebiotic chemistry research and the manuscript by Martinez-Bachs and Rimola addresses one of the most fundamental problems from this topic, i.e. the phosphorylation-induced formation of oligopeptides. The authors use simple cluster calculations to describe the process in gas-phase, clays, as well as in water and disclose the main thermodynamic and kinetic reasons which make this reaction unfeasible in a prebiotic context. I think this is an important piece of information for experimentalists seeking for new alternative ways for amino acid activation plausible on the early Earth. The calculations are performed at an adequate level and the computed data fully support the conclusions drawn by the authors. Therefore, I suggest this fine paper for publication after a minor revision, which should include the following two points:
1. a thorough language editing is recommended: there are a couple of typos in the paper;
2. as it is a custom in quantum chemistry literature, all optimized geometries must be deposited in the supplementary material.
No further review is needed.
Reviewer 2 Report
With the attachment I have indicated that the MS need to be revised extensively both in terms of English, format and style of presenting formulae etc. I have not completed the revision fully, as it is taking too much of my time, since there are many amendments needed to be done. I suggest that this is also gone through by someone else who has a full command of English. HOWEVER, I do feel that MS should not be rejected and I feel that it would appropriate to publish it after extensive revision.

Reviewer 3 Report
An interesting theoretical paper. It should be accepted after some corrections / additions.
At the beginning of the abstract : a sentence should be added about kinetics. Indeed, these reactions arte not only disfavored by thermodynamics but also by kinetics, and this is an important fact.
Are gas phase calculations really relevant? It should be made clear that they are only presented as a background to more realistic results presented afterward.
I'm not sure that phospholipids can be regarded as biopolymers (line 37). They should be listed apart from other examples.
The de novo synthesis of adenosyl monophosphate does not occur as presented in Figure 1B (and in lines 40-43). The authors should refer to the inosine monophosphate synthesis, in which the nitrogen source reacting with an activated ribose is glutamine.
line 58 : "well" before "have" (might well have been).
line 71 "A large number of experiments" (better than "weel suit").
The mechanism shown in Figure 2 is not involved in nowadays biology. An amino acid never reacts with a phosphoric-carboxylic anhydride. These anhydrides react with tRNA's to give esters (or with thiols to give thioesters) so that the studied reaction cannot be regarded as "biology-like". This certainly decreases the interest of the studied reaction (activated Gly + Gly). Authors must mention it clearly and explain why they studied it anyway (which is understandable). In a way the activated Gly + alcohol reaction would have been more relevant. Such a discussion must be presented from the beginning of the paper. As it is, the introduction is an over-simplification of what happens in biology, both for peptides and nucleic acids. It must be rewritten.
Round 2
Reviewer 2 Report
I feel the MS could be tweaked further by a competent grammarian because my previous request was not acted upon. if it had been acted upon, the reader would have picked up words such as solution, intrinsic, bio-inspired and the difference between cations and ions etc. Due to the absence of attention to detail, the MS was made un-necessarily too wordy and complex. I have suggest some improvement and this should be done the letter.
